

# The Brazilian version of the 20-item rapid estimate of adult literacy in medicine and dentistry

Agnes Fátima P. Cruvinel[1], Daniela Alejandra C. Méndez[2], Juliana G. Oliveira[2], Eliézer Gutierres[2], Matheus Lotto[2], Maria Aparecida A.M. Machado[2], Thaís M. Oliveira[2] and Thiago Cruvinel[2]

[1] Department of Public Health/ School of Medicine, Federal University of Fronteira Sul, Chapecó, SC, Brazil
[2] Department of Pediatric Dentistry, Orthodontics and Public Health/ Bauru School of Dentistry, University of São Paulo, Bauru, SP, Brazil

Corresponding author
Thiago Cruvinel,
thiagocruvinel@fob.usp.br

## ABSTRACT

**Background.** The misunderstanding of specific vocabulary may hamper the patient-health provider communication. The 20-item Rapid Estimate Adult Literacy in Medicine and Dentistry (REALMD-20) was constructed to screen patients by their ability in reading medical/dental terminologies in a simple and rapid way. This study aimed to perform the cross-cultural adaptation and validation of this instrument for its application in Brazilian dental patients.

**Methods.** The cross-cultural adaptation was performed through conceptual equivalence, verbatim translation, semantic, item and operational equivalence, and back-translation. After that, 200 participants responded the adapted version of the REALMD-20, the Brazilian version of the Rapid Estimate of Adult Literacy in Dentistry (BREALD-30), ten questions of the Brazilian National Functional Literacy Index (BNFLI), and a questionnaire with socio-demographic and oral health-related questions. Statistical analysis was conducted to assess the reliability and validity of the REALMD-20 ($P < 0.05$).

**Results.** The sample was composed predominantly by women (55.5%) and white/brown (76%) individuals, with an average age of 39.02 years old ($\pm15.28$). The average REALMD-20 score was 17.48 ($\pm2.59$, range 8–20). It displayed a good internal consistency (Cronbach's alpha = 0.789) and test-retest reliability (ICC = 0.73; 95% CI [0.66 − 0.79]). In the exploratory factor analysis, six factors were extracted according to Kaiser's criterion. The factor I (eigenvalue = 4.53) comprised four terms—"*Jaundice*", "*Amalgam*", "*Periodontitis*" and "*Abscess*"—accounted for 25.18% of total variance, while the factor II (eigenvalue = 1.88) comprised other four terms—"*Gingivitis*", "*Instruction*", "*Osteoporosis*" and "*Constipation*"—accounted for 10.46% of total variance. The first four factors accounted for 52.1% of total variance. The REALMD-20 was positively correlated with the BREALD-30 ($Rs = 0.73$, $P < 0.001$) and BNFLI ($Rs = 0.60$, $P < 0.001$). The scores were significantly higher among health professionals, more educated people, and individuals who reported good/excellent oral health conditions, and who sought preventive dental services. Distinctly, REALMD-20 scores were similar between both participants who visited a dentist <1 year ago and ≥1 year. Also, REALMD-20 was a significant predictor of self-reported oral health status in a multivariate logistic regression model, considering socio-demographic and oral health-related confounding variables.

**Conclusion**. The Brazilian version of the REALMD-20 demonstrated adequate psychometric properties for screening dental patients in relation to their recognition of health specific terms. This instrument can contribute to identify individuals with important dental/medical vocabulary limitations in order to improve the health education and outcomes in a person-centered care model.

## INTRODUCTION

Developing countries have a large percentage of people living in areas with deprivation of income and other social rights, such as education and healthcare (*World Bank, The, 2016*). This situation increases the prevalence of adults with difficulties in word recognition, reading, writing, document interpretation, quantitative analysis, communication skills, and conceptual knowledge (*American Medical Association, 1999*; *Berkman et al., 2011*). In 2011, 27% Brazilian people between 15 and 64 year olds were considered functional illiterates, varying from 11% (15–24 year olds) to 52% (50–64 year olds) (*Instituto Paulo Montenegro, 2015*). These limitations hamper the health literacy levels of patients (*Sudore et al., 2006*) and, consequently, their adherence with healthy lifestyle, and their engagement to shared health decision-making process, prevention and treatment of diseases (*Stacey et al., 2017*; *Miller, 2016*; *Lee et al., 2007*; *Friis et al., 2016*).

The health literacy is a powerful predictor of health status in comparison to more distal socio-demographic variables, such as age, income, occupation, education, race and ethnic group (*Schiavo, 2011*; *Bress, 2013*). Low health literacy increases the hospitalization rates, the underuse of preventive services, and the misinterpretation of health information (*Berkman et al., 2011*; *Atchison et al., 2010*), leading to poor oral health conditions, and an inadequate self-perception of dental treatment needs and dental utilization (*Jamieson et al., 2013*; *Holtzman et al., 2014*).

Additionally, individuals with low health literacy express impatience and frustration in using written documents, spending a considerable time filling health forms with incomplete and/or incorrect data (*Schiavo, 2011*). Therefore, clinicians should employ health literacy-based practices in their daily routine focused on improving health outcomes (*Cooper et al., 2003*). Thereby, the availability of instruments to identify individuals with low health literacy is fundamental. Despite the development of several health literacy-based instruments, these tools are more restricted to English-speaking countries, especially to the US (*Altin et al., 2014*; *Parthasarathy et al., 2014*). In Brazil, there are only two validated instruments based on the recognition of words. The Rapid Estimate of Adult Literacy in Dentistry (BREALD-30) (*Junkes et al., 2015*) comprises exclusively dental terms, while the Short Assessment of Health Literacy for Portuguese-Speaking Adults (SAHLPA) (*Apolinario et al., 2012*) presents a list of specific medical terms that need to be interpreted.

Based on the knowledge of the association between medical and dental conditions, such as poor mastication and the impairment of cognitive function (*Tada & Miura, 2017*), adequate periodontal status and glycemic control (*Artese et al., 2015*), the diagnosis of chronic periodontitis and the development of atherosclerotic heart disease (*Dietrich et al., 2017*), and periodontal disease and obesity (*Martens et al., 2017*), the process of health education requires the basic domain of a cross-disciplinary vocabulary by patients, since the misunderstanding of specific terms could hamper the prevention of oral diseases, also impacting negatively the systemic health, and vice-versa (*Gironda et al., 2013*). Hence, it would be desirable the availability of an instrument that evaluates the dynamic interplay of medical/dental words in the patient-health professional communication. In this context, the 20-item Rapid Estimate Adult Literacy in Medicine and Dentistry (REALMD-20) is singular, with the aim of screening individuals by their ability in reading medical and dental terminologies simultaneously, in a simpler and faster way, requiring minimal training of its applicants (*Atchison et al., 2010*; *Gironda et al., 2013*). For instance, the REALMD-20 uses only 20 terms to analyze effectively the recognition of words from two health fields, different from the other tools aforementioned.

Taking into consideration (1) the lack of a Brazilian Portuguese health literacy instrument focused on the simultaneous recognition of dental and medical terms, and (2) the singularity and advantages of the use of REALMD-20 to recognize people with limited health literacy, this study aimed to perform the Brazilian cross-cultural adaptation and validation of this instrument for its application to dental patients in clinical studies.

## MATERIALS & METHODS

This study was previously authorized by the authors of the original REALMD-20 (*Gironda et al., 2013*), and approved by the Human Research Ethics Committee of the Bauru School of Dentistry, University of São Paulo, Brazil (#CAAE 34539714.7.0000.5417), in accordance with the ethical standards of the Declaration of Helsinki.

### Cross-cultural adaptation

The cross-cultural adaptation was performed as described by *Herdman, Fox-Rushby & Badia (1998)* and *Reichenheim & Moraes (2007)*. This process was divided in six distinct steps, as follows: (a) conceptual equivalence, (b) verbatim translation, (c) semantic equivalence, (d) item equivalence, (e) operational equivalence, and (f) back-translation.

#### Conceptual equivalence

Two examiners with expertise in health education analyzed the conceptual framework of the REALMD-20, considering its application adequate to screening Brazilian Portuguese native speakers for limited health literacy levels. The elements analyzed are summarized below:

(1)  the instrument was developed with basis on the concept of health literacy incorporated into The Patient Protection and Affordable Care Act (P.L. 111–148), as follows: "*health literacy is the degree to which individuals have the capacity to obtain, process, and understand basic health information and services needed to make appropriate health*

*decisions*" (*Department of Health and Human Services, 2000*). This definition can be considered appropriate and valid for unlimited cultures;

(2) the REALMD-20 is limited in evaluating the capacity of reading health terms, one aspect of the constellation of abilities needed to construct an adequate health literacy level. Despite this disadvantage, written materials are still the most common source of health information, especially after the advent of the Internet (*Pletneva et al., 2011*). In this context, the relevance of this instrument is supported by its purpose of screening individuals for limited health literacy smoothly, since the ability tested is essential to the acquisition of medical and/or dental knowledge. In addition, the correct reading might indicate indirectly the prior knowledge of the person on specific health terms.

(3) the structure of the REALMD-20 takes into account that the misconception of medical issues could prevent successful dental interventions, and vice-versa. This consideration could add significant data and novel interpretations in clinical and epidemiological studies.

### Verbatim translation, semantic and item equivalences

Initially, the REALMD-20 was literally translated to Brazilian Portuguese by three bilingual health professionals. After the analysis of the independent translations, the semantic of words "*Insurance*" and "*Directed*" seemed non-representative of health-related terms for Brazilian people. Then, they were replaced by equivalent terms in Portuguese, with the aid of a specialist in language and communication. The term "*Insurance*" was replaced by the single word *Covenant* ("*Convênio*") based on its dictionary definition, while the term "*Directed*" was replaced by the word *Instruction* ("*Instrução*"), its synonym cited in the instrument Short Assessment of Health Literacy (S-SAHL) (*Lee et al., 2010*).

### Operational equivalence and back-translation

This stage is crucial to apply the instrument for a reduced sample in a near-real condition, in order to detect possible influences of the instrument's characteristics on the performance of target individuals (*Reichenheim & Moraes, 2007*). The first version of the REALMD-20 consisted of translated/adapted terms that were still ordered similarly to the original version. It was applied for ten adults who attended the dental clinics of the Bauru School of Dentistry, University of São Paulo, Brazil. They received information concerning the aims of the study and signed a statement of informed consent to the inclusion of material pertaining to themselves, with the maintenance of anonymous information and no identification of their acknowledge via the paper.

The influences of the layout and format of the words/instructions, the application setting, and the way of application of the instrument on the results were analyzed in a pre-test by face-to-face interview, when the individuals were asked to report their impression and contributions for improving the application of the tool. This phase was also used as an additional measure to determine the reading difficulty of each word (*Reichenheim & Moraes, 2007*), as described below.

The respondents were predominantly composed by white-brown (70%) and women (80%), with a mean age of 41.8 years and distinct educational levels, varying from primary to tertiary education. They considered the instrument as a simple and easy tool to disclose the

difficulties of patients in understanding health instructions. However, the terms related to *"Abscess"*, *"Calculus"*, *"Allergic"*, *"Instruction"*, *"Constipation"*, *"Extraction"*, *"Jaundice"*, *"Periodontitis"*, *"Amalgam"*, *"Gingivitis"*, and *"Osteoporosis"* were mispronounced by at least one subject. In this scenario, three subjects read two or more listed terms incorrectly. The most reading errors were committed by two subjects with the lowest levels of education (<4 and <8 years of school).

Subsequently, the terms were arranged in order of increasing reading difficulty, considering (a) the number of syllables, (b) the presence of consonant clusters, (c) the word accentuation, (d) the prior knowledge of individuals, and (e) the results obtained during the pre-test. To guarantee the maintenance of cultural and conceptual correspondence between the original and adapted versions, another two independent translators back-translated the instrument to English.

## Validation

Two hundred adults with 18–80 years old from different educational backgrounds were enrolled in this study. They were recruited among patients who attended in the dental clinics of the Bauru School of Dentistry, University of São Paulo, Brazil. Illiterates, non-native Brazilian Portuguese speakers, patients with cognitive, vision, or hearing impairment, and subjects intoxicated by alcohol and/or drugs were not included in the sample. The participants were also informed about the aims of the study and signed a statement of consent. For more details about demographic characteristics of the sample, please see the results.

Six trained investigators asked participants to respond to the Brazilian version of the REALMD-20, the Brazilian version of the Rapid Estimate of Adult Literacy in Dentistry (BREALD-30) (*Junkes et al., 2015*), ten questions from the Brazilian National Functional Literacy Index (BNFLI) (*Junkes, 2013*), and a questionnaire about socio-demographic and oral health-related aspects. The investigators were trained in meeting sessions with basis on a document that described all required actions to the correct application of the instruments. These trainings involved the presentation of diverse situations that were discussed among the participants, such as examples of correct and incorrect readings.

Initially, the participants were invited to take a seat in a comfortable room. They then were interviewed using a questionnaire containing four socio-demographic questions and three oral health-related questions. Following that, they were asked to retain a sheet containing the REALMD-20 (Table 1), and read each one of 20 terms aloud. The words clearly and fluidly pronounced received a score of 1, whereas the inability to read (silence), "trial and error", hesitation of reading, mispronunciation or not attempted words received a score of 0, with an overall score ranging from 0 to 20 (*Gironda et al., 2013*). To determine the test-retest reliability, the REALMD-20 was re-applied to 10% of sample one month later.

Subsequently, the BREALD-30 was applied to evaluate the ability of participants in reading and pronouncing specific dental terms. It comprises 30 words also arranged in order of increasing reading difficulty. The subjects also received one point for each term read correctly, with total scores ranging from 0 to 30 (*Junkes et al., 2015*).
**Table 1  The Brazilian version of the 20-item Rapid Estimate of Adult Literacy in Medicine and Dentistry (REALMD-20).** The English original terms are presented in parentheses.

| REALMD-20 List 1 | List 2 |
|---|---|
| Cárie (*Caries*) | Extração (*Extraction*) |
| Dentadura (*Denture*) | Abscesso (*Abscess*) |
| Higiene (*Hygiene*) | Instrução (*Directed*) |
| Fadiga (*Fatigue*) | Colite (*Colitis*) |
| Anemia (*Anemia*) | Constipação (*Constipation*) |
| Cálculo (*Calculus*) | Osteoporose (*Osteoporosis*) |
| Convênio (*Insurance*) | Gengivite (*Gingivitis*) |
| Alérgico (*Allergic*) | Amálgama (*Amalgam*) |
| Depressão (*Depression*) | Periodontite (*Periodontitis*) |
| Anestesia (*Anesthetic*) | Icterícia (*Jaundice*) |

Finally, the subjects responded to 10 questions of BNFLI. It measures the functional literacy of Brazilian population aged between 15 to 64 year olds (*Junkes, 2013*), by the application of a set of simple tests about the interpretation of figures and documents. In this study, the participants received one point for each right answer, with the overall score varying from 0 to 10.

## Statistical analysis

Data were analyzed using the Statistical Package for the Social Sciences (SPSS) version 21.0 (IBM® SPSS® Statistics, New York, USA).

The exploratory factor analysis (EFA) was conducted to assess the dimensionality of the instrument. The suitability of the dataset for the factor analysis was confirmed by Kaiser–Meyer–Olkin Measure of Sampling (KMO) (>0.60), the Barlett's Test of Sphericity ($P < 0.05$), and the value of the determinant of the correlation matrix (>0.00001). The factors were extracted by principal component analysis (PCA), according to Kaiser's criterion (eigenvalue > 1.0). The Varimax rotation was applied to minimize the number of variables with high loadings in each factor. The items with communalities and factor loadings ≥0.4 were considered acceptable (*Souza et al., 2016*).

The internal consistency and the test-retest reliability of the REALMD-20 were determined by Cronbach's alpha and intraclass correlation coefficient (ICC) for absolute concordance, respectively.

The Spearman's correlation test determined the convergent validity of the REALMD-20 with the BREALD-30 and the BNFLI. The discriminant validity was detected by Mann–Whitney $U$ test through the comparison of REALMD-20 scores between dichotomized socio-demographic variables and oral health-related aspects, as follows: gender (male/female), age (<36 years-old/ ≥36 years-old), race (white-brown/black-asian), education (<12 years/ ≥12 years), occupation (other/health professionals), self-reported oral health (good-excellent/regular-poor), time since last dental visit (<1 year/ ≥1 year), and reason for dental utilization (prevention/treatment).

**Table 2 The distribution of participants by socio-demographic characteristics and oral health-related aspects.**

|  | n (%) |
|---|---|
| **Gender** | |
| Male | 89 (44.5%) |
| Female | 111 (55.5%) |
| **Race** | |
| White-Brown | 171 (85.5%) |
| Black | 28 (14.0%) |
| Asian | 1 (0.5%) |
| **Education** | |
| ≤8th grade | 59 (29.5%) |
| 9–12th grade | 90 (45.0%) |
| College | 37 (18.5%) |
| Post graduate | 14 (7.0%) |
| **Occupation** | |
| Other | 178 (89.0%) |
| Health technician | 6 (3.0%) |
| Health professional | 16 (8.0%) |
| **Self-reported oral health** | |
| Excellent | 21 (10.5%) |
| Very good | 23 (11.5%) |
| Good | 68 (34.0%) |
| Regular | 65 (32.5%) |
| Poor | 21 (10.5%) |
| Do not know/do not answer | 2 (1.0%) |
| **Time since last dental visit** | |
| <1 year | 158 (79.0%) |
| 1–2 years | 20 (10.0%) |
| 2–5 years | 14 (7.0%) |
| >5 years | 7 (3.5%) |
| Do not know/do not answer | 1 (0.5%) |
| **Reason for dental utilization** | |
| Prevention | 68 (34.0%) |
| Treatment | 132 (66.0%) |

The predictive performance of the REALMD-20 for oral health outcomes was analyzed by multivariate logistic regression models, which included only factors with significant Wald statistics in a prior univariate analysis. The factors were ordered into the models with basis on their Wald statistics.

For all analyses, $P$ values <0.05 were considered significant.

## RESULTS

The distribution of socio-demographic characteristics and oral health-related outcomes is presented in Table 2. The sample was composed predominantly by women (55.5%) and

**Table 3** Communalities ($h^2$) and scale means, scale variances and Cronbach's alpha if item deleted from the Brazilian version of the REALMD-20.

| Item | Scale mean if item deleted | Scale variance if item deleted | Cronbach's alpha if item deleted | $h^2$ |
|---|---|---|---|---|
| Caries | 16.49 | 6.64 | 0.791 | 0.58 |
| Denture | 16.48 | 6.61 | 0.789 | 0.93 |
| Hygiene | 16.48 | 6.61 | 0.789 | 0.74 |
| Fatigue | 16.49 | 6.55 | 0.788 | 0.75 |
| Anemia | 16.50 | 6.44 | 0.784 | 0.86 |
| Calculus | 16.51 | 6.17 | 0.774 | 0.76 |
| Covenant | 16.49 | 6.49 | 0.785 | 0.76 |
| Allergic | 16.53 | 6.27 | 0.782 | 0.68 |
| Depression | 16.49 | 6.54 | 0.786 | 0.86 |
| Anesthetic | 16.54 | 6.09 | 0.775 | 0.65 |
| Extraction | 16.50 | 6.44 | 0.784 | 0.73 |
| Abscess | 16.68 | 5.82 | 0.780 | 0.40 |
| Instruction | 16.54 | 6.25 | 0.782 | 0.59 |
| Colitis | 16.56 | 5.99 | 0.773 | 0.64 |
| Constipation | 16.64 | 5.63 | 0.766 | 0.47 |
| Osteoporosis | 16.68 | 5.50 | 0.765 | 0.51 |
| Gingivitis | 16.58 | 5.80 | 0.767 | 0.71 |
| Amalgam | 16.96 | 5.40 | 0.776 | 0.54 |
| Periodontitis | 16.91 | 5.44 | 0.778 | 0.43 |
| Jaundice | 17.02 | 5.46 | 0.779 | 0.61 |

white-brown (85.5%) individuals, with an average age of 39.02 years old ($\pm$15.28, median: 36.50). The average REALMD-20 score was 17.48 ($\pm$2.59, median: 18.00, range 8–20), with the variance of 6.71. If an item was deleted, the scale mean varied between 16.48 and 17.02, while the scale variance ranged from 5.40 to 6.64 (Table 3). The instrument displayed a good internal consistency (Cronbach's alpha = 0.789), with values of Cronbach's alpha if the item was deleted varying between 0.765 and 0.791 (Table 3). The test-retest reliability of the instrument was considered good (ICC = 0.73; 95% CI [0.66–0.79]), with a skewness of $-1.63$ and a kurtosis of 2.90. All participants read the easiest terms "*Denture*", and "*Hygiene*" correctly, whereas only 46% of participants read the most difficult term "*Jaundice*" without errors.

When the algorithm created for the original Rapid Estimate of Adult Literacy in Medicine (REALM) (*Atchison et al., 2010*) was used to categorize these participants in education grade levels, a lower percentage was associated with adequate health literacy. Seventeen (8.5%) participants scored at 4th–6th grade level (range 6–13), 99 (49.5%) participants scored at the 7th to 8th grade level (range 14–18), and 84 (42%) participants scored at high school or more education level.

In the factor analysis, it was observed an adequate sample size (KMO = 0.73) with a non-identity correlation matrix (Bartlett's test of sphericity, $P < 0.001$), and no influence of multicollinearity (determinant = 0.004). Six factors with eigenvalues >1.0

**Table 4  Comparison of average (±SD) and medians of the REALMD-20 scores between dichotomized socio-demographic and oral health-related variables (U-Mann Whitney test, $P < 0.05$).**

|  | Average (±SD) | Median (Min–Max) | P |
|---|---|---|---|
| **Gender** |  |  |  |
| Male | 17.66 ± 2.50 | 18.00 (8–20) |  |
| Female | 17.33 ± 2.66 | 18.00 (8–20) | 0.32 |
| **Race** |  |  |  |
| White-Brown | 17.64 ± 2.51 | 18.00 (8–20) |  |
| Black-Asian | 16.52 ± 2.91 | 17.00 (9–20) | 0.03 |
| **Education** |  |  |  |
| <12 years | 16.94 ± 2.74 | 17.50 (8–20) |  |
| ≥12 years | 19.02 ± 1.16 | 19.00 (16–20) | <0.001 |
| **Occupation** |  |  |  |
| Other professionals | 17.24 ± 2.63 | 18.00 (8–20) |  |
| Health professionals | 19.45 ± 0.74 | 20.00 (18–20) | <0.001 |
| **Self-reported oral health** |  |  |  |
| Good - excellent | 18.34 ± 1.78 | 19.00 (12–20) |  |
| Do not know - regular | 16.39 ± 3.02 | 17.00 (8–20) | <0.001 |
| **Time since last dental visit** |  |  |  |
| <1 year | 17.58 ± 2.56 | 18.00 (8–20) |  |
| ≥1 year | 17.10 ± 2.70 | 18.00 (9–20) | 0.22 |
| **Reason for dental utilization** |  |  |  |
| Prevention | 18.09 ± 2.16 | 19 (10–20) |  |
| Treatment | 17.17 ± 2.74 | 18 (8–20) | 0.01 |

were extracted: the factor I (eigenvalue = 4.53) comprised four terms—"*Jaundice*", "*Amalgam*", "*Periodontitis*" and "*Abscess*"—accounted for 25.18% of total variance, while the factor II (eigenvalue = 1.88) comprised other four terms—"*Gingivitis*", "*Instruction*", "*Osteoporosis*" and "*Constipation*"—accounted for 10.46% of total variance. The first four factors accounted for 52.1% of total variance.

The REALMD-20 was positively correlated with the BREALD-30 ($Rs = 0.73$, $P < 0.001$) and BNFLI ($Rs = 0.60$, $P < 0.001$). The average score (±SD, range) of the BREALD-30 and BNFLI were 24.75 (±4.48, 9–30) and 7.66 (±2.00, 1–10), respectively. The REALMD-20 scores were significantly higher among health professionals, more educated people, individuals who reported good/excellent oral health conditions, and who sought preventive dental services. On the other hand, the scores were similar between both groups of participants who visited a dentist within the last year and those who visited a dentist at ≥1 year (Table 4).

The REALMD-20 was a significant predictor of self-reported oral health status in a multivariate logistic regression model, considering socio-demographic and oral health-related confounding variables (Table 5).

## DISCUSSION

To our knowledge, this is the first validation of the REALMD-20 in a non-English-speaking country. Although focused on the analysis of two skills of health literacy, brief instruments

**Table 5 Logistic regression models for predictive factors of self-reported oral health status and reason for dental utilization.**

|  | B[a] | S.E.[b] | Wald | P | OR[c] |
|---|---|---|---|---|---|
| **Self-reported oral health: good-excellent** | | | | | |
| REALMD-20 | 0.29 | 0.08 | 12.66 | <0.001 | 1.34 |
| Education (≥12 years) | 0.37 | 0.43 | 0.73 | 0.394 | 1.45 |
| Occupation (Health professional) | 1.51 | 0.82 | 3.41 | 0.065 | 4.52 |
| Age | −0.03 | 0.01 | 5.34 | 0.021 | 0.98 |
| Time since last dental visit (<1 y) | 0.86 | 0.40 | 4.57 | 0.033 | 2.36 |
| Constant | −4.73 | 1.47 | 10.30 | 0.001 | 0.01 |
| **Reason for dental utilization: prevention** | | | | | |
| Oral health (good-excellent) | 0.88 | 0.37 | 5.81 | 0.016 | 2.41 |
| Education (≥12 years) | 0.52 | 0.39 | 1.72 | 0.190 | 1.67 |
| Time since last dental visit (<1 y) | 0.99 | 0.47 | 4.56 | 0.033 | 2.70 |
| Occupation (Health professional) | 0.63 | 0.54 | 1.34 | 0.247 | 1.87 |
| REALMD-20 | 0.03 | 0.08 | 0.14 | 0.709 | 1.03 |
| Constant | −2.76 | 1.34 | 4.22 | 0.040 | 0.06 |

**Notes.**
[a] Unstandardized coefficient.
[b] Standard error.
[c] Odds ratio.

as the REALMD-20 are still suitable to rapidly recognize individuals with low health literacy in surveys and clinics (*Gong et al., 2007*). It might support the improvement of health outcomes through the promotion of special education measures for deprived groups (*Miller, 2016*).

This version of the REALMD-20 showed an adequate internal consistency, since Cronbach's alpha >0.7 are more adequate to the analysis of skill performance tools (*Kline, 2000*). However, the Cronbach's alpha was lower than the original REALMD-20. This coefficient varies across populations according to the variation of item prevalences, which render problematic comparisons between different samples (*Teresi & Holmes, 2002*). The theory of reliability is summarized as $1 - (\sigma_e^2/\sigma_x^2)$, where $\sigma_e^2$ is the error variance and $\sigma_x^2$ is the variance of the measure (*Lord & Novick, 1968*). In this scenario, lower reliability estimates are obviously obtained from more homogeneous samples. Therefore, these variations not necessarily reflect true differences in reliability. Here, the variance (6.71) was significantly lower than that described in the original study (10.56) (*Gironda et al., 2013*). It is expected that higher coefficients of reliability will be observed in more heterogeneous population groups, e.g., residents of a city or a country. In this sense, the structure of the twenty items generated after this adaptation process was maintained, supported by the values of communalities and the reduction of scale variance if any item was removed. Additionally, the skewness of this instrument indicates the most normal distribution of data compared to the original REALMD-20 (−2.32), REALM (−4.01) and REALM-D (−3.84) (*Atchison et al., 2010*; *Gironda et al., 2013*), which contribute with its capacity of discrimination. Six subscales were evidenced by EFA, with the detection of a major factor (2.41-fold greater

than factor II), composed by the terms with the highest reading difficulty, except for "*Abscess*". Therefore, health literacy measured by the REALMD-20 is multidimensional.

The mean score observed in this study (17.48) was negligibly higher than that previously described by *Gironda et al. (2013)* (17.28), especially when considering the same median value (18.0) achieved by both studies. Interestingly, most of these participants (74.5%) attended the school for a shorter time than the expectancy for Brazilians (15 years) (*Central Intelligence Agency, 2017*). Then, the mean difference could be explained by distinct socio-demographic characteristics between the participants from each validation, although both groups of volunteers were recruited in dental clinics. The present sample was composed exclusively by Brazilian Portuguese adult native speakers, with a median age (36.50) close to that for total Brazilian population (31.60). Also, the distribution of age groups was similar to that found in Brazil, with 21.5% *vs.* 15.3% (18–24 *y*), 61.0% *vs.* 56.8% (25–54 *y*), 11.0% *vs.* 11.5% (55–64 *y*), and 6.5% *vs.* 10.4% (65+ *y*) (*Central Intelligence Agency, 2017*), corroborating with no influence of age groups on the results of this validation. It was demonstrated that health literacy progressively decreases toward older age groups, young adults being more prone to obtain health information when engaged in health programs (*Neter & Brainin, 2012*). Likewise, the use of electronic sources as the Internet, combined with the accessibility of information, contributes to greater health literacy levels among young adults (*Neter & Brainin, 2012*; *Sun et al., 2013*). These present results indicate a greater proportion (58%) of dental patients with limited health literacy in comparison with the original study (52.5%) (*Gironda et al., 2013*).

The dichotomized categories of educational levels and races were significantly associated with the capacity of reading health terms. As expected, health professionals presented higher REALMD-20 scores than other workers, which indicate the association of these outcomes with health knowledge levels. These findings are consistent with other prior studies (*Lee et al., 2007*; *Atchison et al., 2010*; *Gironda et al., 2013*; *Sun et al., 2013*). The convergent validity of the REALMD-20 was performed with an oral health literacy tool (BREALD-30) ($Rs = 0.73$) and an instrument for analyzing the functional literacy of Brazilian citizens (BNFLI) ($Rs = 0.60$). The lack of a validated word recognition-based tool impeded this convergent analysis for medical terms. The most similar instrument in this field, SAHLPA (*Apolinario et al., 2012*), is marked by a distinct framework that requires the recognition and interpretation of words.

The participants with good/excellent self-reported oral health status and users of preventive dental services achieved higher REALMD-20 scores. In addition, health literacy and the type of occupation were significant predictors of self-reported oral health status in a multiple logistic regression model. Each one-point increase in the score of REALMD-20 indicates 34% more chance of an individual self-reporting a good-excellent oral health. These findings are supported by the previous associations of low health literacy with poor oral health, more dental treatment needs and overuse of emergency services (*Burgette et al., 2016*). On the other hand, REALMD-20 scores were not associated with the periodicity of dental visits. The participants were dichotomized in <1 year and ≥1 year from their last dental visit, considering that 79% of them reported regular dental visits. Although our results are in agreement with other authors (*Jamieson et al., 2013*;

*Gong et al., 2007*; *Griffey et al., 2014*), this issue presents divergent outcomes. *Calvasina et al. (2016)* showed that Brazilian immigrants who attended dental care sporadically were nearly five times more likely to have inadequate oral health literacy than those who visited a dentist annually or more often. It is noteworthy that our results could be influenced by the inaccuracy of information provided by participants during the collection of oral health-related information, which would be justified by memory errors and social desirability biases, i.e., when individuals deny some undesirable traits. This hypothesis gains strength if we consider that dentists were responsible for interviewing the participants.

This study presents some limitations. First, as commonly observed in other validation studies (*Lee et al., 2007*; *Atchison et al., 2010*; *Junkes et al., 2015*; *Apolinario et al., 2012*; *Gironda et al., 2013*; *Lee et al., 2010*; *Souza et al., 2016*; *Gong et al., 2007*), the REALMD-20 was applied to a convenience sample that does not reflect the genuine structure of Brazilian demographics. It is inherent of this methodological approach that excluded illiterates (7.4% of total population) (*Central Intelligence Agency, 2017*), and included a higher percentage of health professionals (11%) in relation to official statistics (1.4%) (*World Health Organization, 2011*). This design is supported by two main reasons: (a) the restriction of including only potential word readers, and (b) the need of analysis of discriminant validity between health professionals and other workers. Second, the proportion between males and females in adult ages was lower than that expected (0.80 *vs.* 0.94), which could be explained by the recruitment of participants in dental clinics, since women are more interested in seeking health care (*Thompson et al., 2016*). Third, the gratuity of dental services offered by our public institution probably attracts a great percentage of patients living in areas of more vulnerable social conditions, which corroborate with the lower proportion of white-brown participants (85.5%) in relation to Brazilian demographics (90.8%) (*Central Intelligence Agency, 2017*). Fourth, the great number of investigators increases the risk of miscellaneous judgments on correct and incorrect readings. Although this possibility cannot be discarded, all efforts were directed to standardize the decision-making process of all investigators, with previous discussion and training. Fifth, although this instrument presented a good stability (ICC = 0.73), its potential for detecting people with low health literacy levels might be hampered in longitudinal studies, because the prior knowledge of words could facilitate the reading over time. In this sense, the effect of previous awareness of words might influence this analysis of stability, contributing to the decrease of ICC values. However, in our opinion this situation was minimized by the observation of an interval of one month between two applications. Finally, the predictive validation was established only on oral health-related outcomes. In theory, general health conditions are also linked to health literacy levels; hence, further studies must be developed to confirm the predictive value of this tool for systemic findings.

In conclusion, this Brazilian version of the REALMD-20 demonstrated adequate psychometric properties for screening dental patients in relation to their recognition of health specific terms, contributing to identify individuals with dental/medical vocabulary limitations for the improvement of health education and outcomes in a person-centered care model. Further studies need to be carried out in order to validate the REALMD-20 for its application in distinct population groups.

### Funding

This work was supported by the São Paulo Research Foundation (grant #2014/21515-1). The funders had no role in study design, data collection and analysis, decision to publish, or preparation of the manuscript.

### Grant Disclosures

The following grant information was disclosed by the authors:
São Paulo Research Foundation: #2014/21515-1.

### Competing Interests

The authors declare there are no competing interests.

### Author Contributions

- Agnes Fátima P. Cruvinel conceived and designed the experiments, performed the experiments, analyzed the data, contributed reagents/materials/analysis tools, wrote the paper, prepared figures and/or tables, reviewed drafts of the paper.
- Daniela Alejandra C. Méndez, Juliana G. Oliveira, Eliézer Gutierres, and Matheus Lotto performed the experiments, reviewed drafts of the paper.
- Maria Aparecida A.M Machado analyzed the data, contributed reagents/materials/analysis tools, reviewed drafts of the paper.
- Thaís M. Oliveira analyzed the data, contributed reagents/materials/analysis tools, reviewed drafts of the paper, translation from English into Brazilian Portuguese.
- Thiago Cruvinel conceived and designed the experiments, analyzed the data, contributed reagents/materials/analysis tools, wrote the paper, prepared figures and/or tables, reviewed drafts of the paper.

### Human Ethics

The following information was supplied relating to ethical approvals (i.e., approving body and any reference numbers):

This study was approved by the Human Research Ethics Committee of the Bauru School of Dentistry, University of São Paulo (#CAAE 34539714.7.0000.5417).

### Data Availability

The raw data has been supplied as a Supplemental File.

### Supplemental Information

Supplemental information for this article can be found online at http://dx.doi.org/10.7717/peerj.3744#supplemental-information.

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
