# Peer review of "The Brazilian version of the 20-item rapid estimate of adult literacy in medicine and dentistry"

_PeerJ, doi:10.7717/peerj.3744_

## Round 0.1 · original submission · Major Revisions

In my opinion, the rationale for this study is not clearly stated, which is also something both reviewers are commenting on below. Major revisions are needed on the experimental design description and data interpretation and discussion areas of the manuscript. Please revise your manuscript including specific details about how the participants were recruited and details about participants' demographics. These details need to also be extensively discussed in the manuscript, as one of the weakest points of the manuscript is the fact that your high educational level sample might not necessarily reflect the Brazilian population, and therefore more studies would be needed to fully validate this Brazilian version instrument. If re-sampling is not possible, please limit your conclusions and remove the word validation from your title. In addition to these major points, please address the reviewers comments.

Reviewer 1 ·

Basic reporting

Introduction - line 21, page 3: Reference [16] is not the best reference for this information. Please consider:
- Altin SV, Finke I, Kautz-Freimuth S, Stock S. The evolution of health literacy assessment tools: a systematic review. BMC Public Health. 2014;14:1207. doi: 10.1186/1471-2458-14- 1207.
- Parthasarathy DS, McGrath CP, Bridges SM, Wong HM, Yiu CK, Au TK. Efficacy of instruments measuring oral health literacy: a systematic review. Oral Health Prev Dent. 2014;12(3):201-207.

Experimental design

Introduction - lines 31-32, page 3: The authors should improve the argumentation to better justify the study.
Material and Methods
1) line 13, page 5: Please demographically characterize the participants of the Operational Equivalence (age, race, socioeconomic level and schooling, gender, place where they were accessed ...).
2) Was the ability to discriminate different levels of health literacy of the translated instrument assessed at this stage? (Operational Equivalence, lines 12-25, page 5)
3) Please clarify the setting where the study participants (Validation) were selected and who were them (line 28, page 5).
4) Can the authors provide more detail about how the researchers were trained? Do you think that having had so many examiners applying the instruments may have introduced some sort of bias to data? (line 3, page 6)
5) I also encourage the authors to provide more details - type and response options- about the questions in lines 21-22, page 6.
6) Why did the authors choose 2 y as the cutoff point for the variable 'time since last dental visit'? (line 13, page 7) See variable distribution in Table 2.
7) Please clarify which type of regression was adopted in the multivariate analysis. The text states in line 14 page 7 'linear regression models', but the outcomes were dichotomous and this type of regression presupposes a numerical outcome.

Validity of the findings

1) Lines 10-11, page 6: Since it requires a second reading, is test-retest an adequate check for this type of instrument? Is it not possible that a repeated reading of the instrument leads to a change in the individual's ability to read and pronounce the words? I would like to see a discussion on this.
2) All validity types have been tested only for oral health literacy and not for non-specific health literacy. Please explain why you did not use the SAHLPA and other medical indicators.
3) The high scores of the REALMD-20 (with low variance) obtained by the participants (Results: line 23 page 7, Discussion: lines 5-12 page 9) induces some important questions. Was the literal translation of the instrument sufficient to ensure its power of discrimination? How is it possible assess the ability to discriminate patients with distinct educational backgrounds, as proposed by the authors in lines 7-8 page 2, or to screen for limited dental/medical health literacy, as stated by the authors of the original REALMD-20 (Gironda et al 2013), with a high educational level sample (see Table 2)? Does the sample used reflect Brazilian population? Is it consistent with the objectives of the proposed instrument?
4) Table 4: Since a non-parametric test was used (Lines 8-9, page 7, Mann-Whitney U test), I suggest the authors to include non-parametric measures on the Table, as well as the test used (Mann-Whitney U test).
5) The statement 'These results are in agreement with the original study' is rather vague (line 30 page 8). The authors should discuss the fact that the Cronbach’s alpha coefficient observed in this study have been lower than that previously described by Gironda et al. (2013) and Atchison et al. (2010).
6) Why the assumption of unidimensionality of the instrument (lines 27-28 page 6 and lines 3-4 page 4), if the authors of the original instrument have found 2 main factors?
7) The proposed explanation for the lack of association between OHL and 'time since last dental visit' is not clear and does not add relevant information as it is. (lines 24-28, page 9)
8) To conclude that 'this version of the REALMD-20 demonstrated adequate psychometric properties to screening native Brazilian adults for health literacy in epidemiological studies and clinical environments' (lines 1-4, page 10), it is necessary to better explain the study setting.

Additional comments

I appreciated the opportunity to review this manuscript. It was mostly well written and reported a process, which followed standards, of translation, cultural adaptation and validation of an instrument (REALMD-20) to assess the health literacy (HL) of Brazilian adults in a sample of 200 individuals. Until now, there are few health literacy assessment tools available in Brazilian Portuguese. With that said, I think that the study needs revision before the acceptance (as I have noted above). Also, some minor suggestions:
1) Abstract - line 11, page 2: The adaptation process was not cited. Please characterize the study participants.
2) Keywords: I suggest that the authors include 'literacy in dentistry' and 'psychometric evaluation'
3) Not all references are available in English [4, 25, 27]. Please re-write the references titles in English to ensure that the international audience can understand them.

·

Basic reporting

1) I would suggest that the authors submit the manuscript for Academic English proofreading. Although the paper is overall well -written there are a few grammar inadequacies, especially with the use of the pronoun "The". For instance: Is it "The Health Literacy"or Health Literacy" (line 88). I would review the whole paragraph as maybe "enlarges" is also not the best word to use (line 92).

2) Please add "Brazilian" before National Functional Literacy instruments to clarify to the audience that this tool was developed in Brazil. This should be consistent in the entire paper.

3) Could you add the exact p-values in Table 4?

4) It is not clear to me what is the difference between REALD-20 and REALMD-20 ? For instance, in the dental practice, which one is more appropriate? In other words, I suggest that the authors strengthen their rationale on why is it important to validate the REALMD-20? What is the application of that instrument? Why is it different from REALD-20, in which ways? Is it better? Please be more specific.

Experimental design

Major corrections:

1) How the participants were recruited? Where? This information should be included up in the front in the methods section. Also you included elderly in your sample. How this option impacted your results?

Lines 168-169: "Two hundred adults with 18-80 years old from different educational backgrounds were enrolled in this study".

Validity of the findings

2) The results showed that (line, 246):

"Individuals who visited a dentist within two last years presented similar health literacy levels to those who visited a dentist at > 2 years (P=0.083)". Where is the Table showing this result? Is it Table 4? Are you considering a p-value significant of less than 0.05? This is a very important issue in your paper, and it is not clearly stated.

Also, if this result is significant, the explanations you provided in the discussion are not clear either. Other Health Literacy studies with a Brazilian population, although using different instruments, have shown an association between low levels of HL and low participation in the oral health care process, including low dental utilization (Calvasina et al., 2016). This matter should be clarified and supported by literature.

3) The manuscript needs a more in depth discussion on the main limitations.

Additional comments

I commend the authors for adequately following the proper methodological steps to cross-culturally validate such important instrument into the Brazilian context. However, one of the main concerns I have with this manuscript regards how the participants were recruited and how their socio-demographics reflect the overall Brazilian population, which you mentioned, comprised of "In 2011, 27% 84 Brazilian people between 15 and 64 year olds were considered functional illiterates, varying from 85 11% (15-24 year olds) to 52% (50-64 year olds) [4]" (lines 84-85). Apparently, your sample is comprised of well-educated Brazilian citizens, as your results showed that the majority of the sample had a good HL (average 17). (Also, this should be discussed in your results).

---

## Round 0.2 · Minor Revisions

Thank you for addressing reviewers comments and concerns. In my opinion the manuscript is almost ready for publication. Please address the reviewer comments and send it back as soon as you can.

Reviewer 1 ·

Basic reporting

All topics were satisfactory addressed.

Experimental design

All topics were satisfactory addressed.

Validity of the findings

All topics were satisfactory addressed.

Additional comments

I congratulate the authors for doing a good job in answering satisfactory the topics addressed, so I consider that the revised manuscript is ready to be published into the periodic Peer J and to give an important contribution to researches on health literacy.

Following are 2 comments on minor revisions.

1) About the assumption of unidimensionality of the instrument, I still disagree with the authors since previous studies have already demonstrated that health literacy are not unidimensional. I suggest they adapt the text to something similar to:
"The exploratory factor analysis (EFA) was conducted to assess the dimensionality of the instrument. "(page 8, lines 3-4; and also adapt lines 18-19 of page 10).

2) The following excerpt from the discussion section is not clear: "On the other hand, when the algorithm created for the original Rapid Estimate of Adult Literacy in Medicine (REALM) [12] was used to categorize these participants in education grade levels, a lower percentage was associated with adequate health literacy. Seventeen (8.5%) participants scored at 4th-6th grade level (range 6-13), 99 (49.5%) participants scored at the 7th to 8th grade level (range 14-18), and 84 (42%) participants scored at high school or more education level." Are these results from this study or from Gironda et al.? Where can the readers see them in the results?

---

## Round 0.3 · accepted · Accept

Congratulations to the authors for addressing all the suggested modifications and clarifications.